# The Additional Medical Expenditure Caused by Depressive Symptoms among Middle-Aged and Elderly Patients with Chronic Lung Diseases in China

**DOI:** 10.3390/ijerph19137849

**Published:** 2022-06-26

**Authors:** Sihui Jin, Yun Wu, Shengliang Chen, Dongbao Zhao, Jianwei Guo, Lijin Chen, Yixiang Huang

**Affiliations:** Department of Health Policy and Management, School of Public Health, Sun Yat-sen University, 74 Zhongshan 2nd Road, Guangzhou 510030, China; jinsh7@mail2.sysu.edu.cn (S.J.); wuyun33@mail2.sysu.edu.cn (Y.W.); chenshliang@mail2.sysu.edu.cn (S.C.); zhaodb3@mail2.sysu.edu.cn (D.Z.); guojw8@mail2.sysu.edu.cn (J.G.); chenlijin@mail.sysu.edu.cn (L.C.)

**Keywords:** chronic lung diseases, depressive symptoms, economic burden, Chinese middle-aged and elderly

## Abstract

Depression is one of the most common comorbidities in patients with chronic lung diseases (CLDs). Depressive symptoms have an obvious influence on the health function, treatment, and management of CLD patients. In order to investigate the additional medical expenditure caused by depressive symptoms among middle-aged and elderly patients with CLDs in China, and to estimate urban–rural differences in additional medical expenditure, our study used data from the 2018 China Health and Retirement Longitudinal Study (CHARLS) investigation. A total of 1834 middle-aged and elderly CLD patients were included in this study. A generalized linear regression model was used to analyze the additional medical expenditure on depressive symptoms in CLD patients. The results show that depressive symptoms were associated with an increase in medical costs in patients with CLDs. Nevertheless, the incremental medical costs differed between urban and rural patients. In urban and rural patients with more severe comorbid CLD and depressive symptoms (co-MCDs), the total additional medical costs reached 4704.00 Chinese Yuan (CNY) (USD 711.60) and CNY 2140.20 (USD 323.80), respectively. Likewise, for patients with lower severity co-MCDs, the total additional medical costs of urban patients were higher than those of rural patients (CNY 4908.10 vs. CNY 1169.90) (USD 742.50 vs. USD 176.90). Depressive symptoms were associated with increased medical utilization and expenditure among CLD patients, which varies between urban and rural areas. This study highlights the importance of mental health care for patients with CLDs.

## 1. Introduction

Chronic lung diseases (CLDs) are the third leading cause of death both in China and worldwide, and nearly 7.4% of the world’s population currently live with CLDs, making it a major contributor to the global burden of non-communicable diseases [1,2]. The main categories of CLDs include chronic obstructive pulmonary disease (COPD), chronic bronchitis, emphysema, and other chronic respiratory diseases. Among these, COPD is characterized by progressive and irreversible airway obstruction, and it accounts for the majority of CLD-attributable deaths and disability-adjusted life years [3]. In 2017, there were more than 300 million new cases of COPD worldwide, making it the seventh leading cause of disability [4]. With the number of cases increasing, COPD was third among the leading causes of death globally in 2019 [5]. Numerous studies have found several major extra-pulmonary effects and significant comorbidities, which may exacerbate the severity of the disease in CLD patients [6,7,8,9]. Depression was recognized as one of the most common comorbidities of CLDs [10]. The prevalence of both anxiety and depression may even be higher among COPD patients compared with other chronic diseases [11]. In 2012, the prevalence of depression among COPD patients in China was about 36% [12]. A meta-analysis study [13] found that the risk of depression in COPD patients was 1.69 times higher than in the general population. In China, patients with COPD were 1.9 times more likely to experience depression compared to non-COPD patients [14]. Additionally, several studies have demonstrated that the risk of depression is higher in COPD patients when compared with the general population, even after adjusting for severity of disease [15,16,17].

In CLD patients, depressive symptoms can complicate medication regimens and negatively affect medication compliance; these problems lead to increased health care utilization and expenditure [18,19,20,21,22]. A study based on the IMS Life Link Database in the USA showed that direct medical costs for COPD patients with depressive symptoms were approximately 1.2 times higher than those for patients without depressive symptoms [23]. A study conducted in Maryland [24] also found significantly higher health care utilization and medical costs in patients with depression-comorbid chronic lung disease compared to those with non-comorbid CLDs and those with depressive symptoms alone (non-co-MCDs). A population-based cohort study in Canada found that the average hospital costs for patients with COPD and depression were CAD 16,749.40 (USD 13,166.70) higher than for depressed patients without COPD [25]. However, there are no studies on the effect of depressive symptoms on the additional medical costs of patients with comorbid chronic lung diseases and depressive symptoms in China.

Further studies have suggested that the impact of depressive symptoms on COPD patients’ costs may be multifactorial, covering gender, age, address, smoking status, and disease-related factors (e.g., long-term oxygen therapy and walking ability) [26,27,28]. A number of studies have compared the differences between rural and urban residents in the utilization of medical care services, and these studies have shown that urban or rural residence status can affect health care utilization and costs [29,30]. Research has generally found lower utilization of all sorts of health care services in rural areas [31,32,33,34,35,36] due to the unbalanced development of medical services between urban and rural areas. Studies in China have consistently reported that rural older adults have higher levels of depressive symptoms than the urban population [37,38]. As for health care, the hospitalization costs, length of stay, and frequency of inpatients in rural areas were lower than in urban areas [39]. However, compared with CLD patients living in urban areas, rural CLD patients experienced poorer health outcomes and higher rates of hospitalization and mortality [40].

As the number of CLD patients with comorbid depressive symptoms increases, more and more studies have focused on the association between depressive symptoms and CLDs [41]. Previous studies have proved that depressive symptoms can affect the treatment of CLDs patients and exacerbate their conditions [42,43]. However, limited information is available on the effects on the medical economic burden created by depressive symptoms in CLD patients, particularly regarding differences between urban and rural status. The objectives of this study were to evaluate the additional medical utilization and expenditure caused by depressive symptoms in middle-aged and elderly people with CLDs in China, and to estimate urban–rural differences in additional medical expenditure. Consequently, this study put forward the following two hypotheses.

**Hypothesis** **1.***Depressive symptoms are associated with additional medical expenditure and utilization in middle-aged and elderly people in China with chronic lung diseases*.

**Hypothesis** **2.***The additional medical expenditure and utilization caused by depressive symptoms among middle-aged and elderly chronic lung disease patients in China are different in urban and rural areas*.

## 2. Materials and Methods

### 2.1. Data Sources

Our study used the data derived from the 2018 China Health and Retirement Longitudinal Study (CHARLS) database. CHARLS was a survey of middle-aged and elderly people in China, based on a sample of households with members aged 45 years or above. It aimed to establish a high-quality public micro-database to provide a wide range of information, from socioeconomic status to health conditions, to meet the needs of scientific research on middle-aged and elderly people [44]. CHARLS followed the top-down county-neighborhood-household-individual order procedure, using the Probabilities Proportional to Size (PPS) sampling method [45]. CHARLS respondents were followed every 2 years using a face-to-face computer-assisted personal interview (CAPI).

### 2.2. Criteria for Depressive Symptoms

The 10-item Center for Epidemiological Studies Depression Scale (CESD-10), a modified version of the 20-item CESD scale, was used to assess depressive symptoms [46]. Respondents were asked to rate their response to the question, “how often have you felt this way in the past week?” on a scale of 0 to 30. We used a universal criterion (score ≥ 10) to identify individuals with significant depressive symptoms. The CESD-10 exhibited good internal consistency in the general population, as well as reliability and validity for the elderly in Chinese communities; the Cronbach’s coefficient of every CHARLS investigation was greater than 0.75 [47]. In order to observe the disparity in economic burden effects caused by different degrees of depression among CLD patients, our study divided depressive symptoms into two levels, defined as mild to moderate depressive symptoms (with a score of 10 ≤ X ≤ 20) and severe depressive symptoms (with a score of 21 ≤ X ≤ 30). These classifications were used to identify a group of patients with lower severity comorbid CLD and depressive symptoms (lower co-MCDs) and a group with higher severity co-MCDs (higher co-MCDs).

### 2.3. Study Sample

Our research used the data from the fourth CHARLS survey, with 19,528 individuals investigated in total. Whether people had a CLD was judged according to questions (e.g., “Have you been diagnosed with chronic lung diseases, such as chronic bronchitis, emphysema (excluding tumors, or cancer) by a doctor?”). There were 11,710 non-CLD patients and 972 people who declined to be questioned or did not complete the survey. Finally, our study included 1834 participants, including 1122 (61.2%) patients with non-comorbid CLDs, 570 (31.1%) with lower co-MCDs, and 142 (7.7%) with higher co-MCDs.

### 2.4. Medical Utilization and Expenditures

In this study, we assessed the visit times and costs of outpatient care, inpatient care, and self-medication. The visits and costs of outpatient care and self-medication for patients in a year were calculated by multiplying by 12 the amount self-reported in the latest month. The inpatient visits and costs were calculated directly based on the self-reported amount in the past year. Costs were presented in Chinese Yuan (CNY), and in 2018 the official currency conversion rate was CNY 6.61 per USD 1.00.

### 2.5. Research Variables

In our study, we used the medical costs and visits of middle-aged and elderly CLD patients as the outcome variables, including outpatient visits and costs, inpatient visits and costs, and self-medication visits and costs. Other research variables included gender, age (categorized into age groups of 45–54, 55–64, 65–74, and ≥75 years), rural–urban status, marital status, and education level (categorized into primary school and below, middle school, and high school and above), smoking status, type of medical insurance, occupations (categorized into unemployed/unpaid, retired, non-agricultural work, and agricultural work), and household monthly consumption levels (averaged into 5 levels).

### 2.6. Statistical Analysis

The statistical differences in the demographic characteristics between patients with non-co-MCDs, lower co-MCDs, and higher co-MCDs were tested by the Chi-square test. The null hypothesis specified that there were no differences in demographic characteristics among the three groups. Considering that there were a large number of zero values for visit times, we used a zero-inflated negative binomial model to measure the relationships between each variable and the number of visits. We reported the incidence rate ratio (IRR) and 95% CI to compare the influence of depressive symptom severity and other variables on total visit counts. IRR was calculated as the ratio of the total number of medical visits of patients with a characteristic variable to those in the reference group. Then, according to the data dispersion degree, we adopted multiple analyses. The specific procedures are as follows. Firstly, we used a zero-inflated negative binomial regression model to analyze the total number of health care visits and outpatient visits. With probability π, the response of the first process was a zero count, and with a probability of (1−π), the response of the second process was governed by a negative binomial with mean *λ*, which also generated zero counts. The overall probability of zero counts was the combined probability of zeros from the two processes. Thus, a zero-inflated negative binomial regression model for the response *Y* can be written as:$$P\left(Y=0\right)=\pi +\left(1-\pi \right){\left(1+k\lambda \right)}^{{-}_{k}^{1}}$$
$$P\left(Y=y\right)=\left(1-\pi \right)\frac{\tau \left(y+\frac{1}{k}\right){\left(k\lambda \right)}^{y}}{\tau \left(y+1\right)\tau \left(\frac{1}{k}\right){\left(1+k\lambda \right)}^{y+\frac{1}{k}}}$$
where y=1, 2, …. Secondly, to evaluate the inpatient visits and self-medication visits, a zero-inflated Poisson regression model was used that included a logit model for predicting excess zeros and a Poisson count model. The equation can be expressed as:P(Y=0)=π+(1−π)exp(−λ)
$$P\left(Y=y\right)=\left(1-\pi \right)\frac{\exp\left(-\lambda \right)\lambda^{y}}{y!}$$
where y=1, 2, …. Due to the characteristics of consecutiveness and positive skewness in the health care costs, we selected a generalized linear model (GLM) with log-link and gamma distribution to evaluate the incremental medical costs. All estimated values used robust regression results controlled by key covariates including gender, age, address, marital status, education level, smoking status, type of medical insurance, occupation, and household monthly consumption levels. In addition, all results were presented by predictive margin effects. The statistical significance *p*-value was set to 0.05 using two-tailed tests. Analyses of data were conducted using Stata version 16.0 (StataCorp, College Station, TX, USA).

## 3. Results

### 3.1. Prevalence of Depressive Symptoms and Characteristics of the Study Sample

The prevalence of depressive symptoms in middle-aged and elderly CLD patients in China was 38.8%, with 31.1% lower co-MCD patients and 7.7% higher co-MCD patients (see Table 1). Compared with non-co-MCD patients, co-MCD patients were more likely to be female (*p* < 0.001), be 55–74 years old (*p* < 0.001), live in a rural area (*p* = 0.024), be separated/divorced/unmarried (*p* = 0.016), have an education level of primary school and below (*p* < 0.001), be a smoker (*p* = 0.019) and be unemployed (*p* = 0.015) (Table 1).

### 3.2. Model Results

The incidence rate ratio (IRR) of medical visits, based on the total number of medical visits, was estimated for each variable in the model (Table 2). Variables having a significant impact on medical visits included depressive symptom status (IRR = 1.30 and 1.29, *p* < 0.001), living in a rural area (IRR = 1.13, *p* < 0.05), being retired (IRR = 0.81, *p* < 0.05), employment in agricultural work (IRR = 0.97, *p* < 0.05), having a household monthly consumption expenditure level in the lower 20% or the highest 20% (IRR = 0.85, IRR = 0.85, respectively, *p* < 0.05 for both); there was no statistical significance for the other variables.

### 3.3. Estimating the Additional Medical Utilization and Expenditure Caused by Depressive Symptoms

Depressive symptoms were associated with additional medical visits and expenditure in middle-aged and elderly patients with co-MCDs. For urban patients, the additional costs of lower co-MCD patients reached CNY 4908.20 (USD 742.50) (*p* < 0.05), and the additional costs for higher co-MCD patients were CNY 4704.00 (USD 711.60) (*p* < 0.001). For outpatient services, the additional costs of higher co-MCD patients were twice as much as those of lower co-MCD patients (CNY 812.30 vs. CNY 1651.20) (USD 122.90 vs. USD 249.80). In addition, the incremental inpatient costs of co-MCD patients were CNY 2194.60 (USD 332.00) and CNY 2083.70 (USD 315.20), respectively, for higher and lower co-MCD patients. For rural patients, the number of additional medical visits of lower co-MCD patients was higher than for higher co-MCD patients (2.5 vs. 1.9, *p* < 0.05 for both), and the total additional costs of lower co-MCD patients were less than those of higher co-MCD patients (CNY 1169.90 vs. CNY 2140.20, *p* < 0.05 for both) (USD 176.90 vs. USD 323.80). However, there were no statistically significant additional outpatient costs in either of the two groups. The incremental inpatient costs of lower co-MCD patients were CNY 690.90 (USD 104.50) (*p* < 0.05), and those of higher co-MCD patients reached CNY 1122.60 (USD 169.80) (*p* < 0.001) (Table 3).

## 4. Discussion

Our study found that depressive symptoms were significantly associated with higher health care utilization and expenditure among CLD patients. Based on the average economic loss for non-co-MCD patients of CNY 8375.20 (USD 1267.00), the additional medical costs caused by depressive symptoms increased by 20.0% in patients with lower co-MCDs and 69.2% in higher co-MCD patients. Results from the National Emphysema Treatment Trial [48] in the United Kingdom revealed that depression increases patient hospitalizations due to respiratory conditions and COPD. In addition, a study of the IMS Life Link database in the USA [23] showed that patients with COPD and comorbid depression were 60% more likely to have hospitalization visits compared to COPD patients without depression, and their annual direct medical costs were also higher (USD 3185.00 vs. USD 2680.00). The Maryland study [24] based on COPD patients also showed that depression increased the annual average number of medical visits by 17.7, and the corresponding medical costs were USD 9057.00, higher than those of COPD patients without depression (USD 7725.00). These results are consistent with our findings based on the Chinese population. On the one hand, the depressive symptoms of elderly CLD patients may be covered by physical symptoms. The physical discomfort caused by depression may prompt CLD patients to seek medical care, resulting in an increase in outpatient visits and hospitalization frequency [49]. On the other hand, depression-related feelings, such as hopelessness, may prevent co-MCD patients from reporting their somatic symptoms immediately, resulting in increased disease severity and higher medical costs [50]. In addition, insufficient mental health resources and the lack of integration of psychiatry and general medicine also impede patients’ recovery and increase medical service utilization [51,52].

Our study also found that the severity of depressive symptoms was significantly associated with a higher financial burden for CLD patients. For rural patients, the additional medical costs of higher co-MCD patients were twice as high as those of lower co-MCD patients (CNY 2140.20 vs. CNY 1169.90) (USD 323.80 vs. USD 176.90). Urban patients also showed an increase in additional outpatient costs as depressive symptoms exacerbated (CNY 812.30 vs. CNY 1651.20) (USD 122.90 vs. USD 249.80). A previous cross-sectional study of older German adults reported that higher total health care costs were associated with moderate to severe depressive symptoms [53]. A study in the United States [54] also suggested that the financial burden of illness increased as depressive symptoms worsened, and the increased medical costs of major depression patients were about 1.15 times those of patients with mild depression. It is known that major depressive symptoms in old age are often accompanied by symptoms such as fatigue, dizziness, headache, abdominal pain, and back pain. These phenomena may lead to the increased use of treatment-related medical and pharmaceutical services by patients with severe depressive symptoms [55].

In addition, we found urban–rural differences in the medical costs of co-MCD patients. For lower co-MCD patients, the additional medical costs of urban patients were nearly four times higher than those of rural patients (CNY 4908.10 vs. CNY 1169.90) (USD 742.60 vs. USD 176.90). In addition, for higher co-MCD patients, the additional medical costs of urban patients were nearly twice those of rural patients (CNY 4704.00 vs. CNY 2140.20) (USD 711.60 vs. USD 323.80). Moreover, patients with depressive symptoms may be hospitalized for physical symptoms [56]. As depressive symptoms worsen, additional hospitalization costs nearly tripled for rural co-MCD patients (CNY 690.90 vs. CNY 1122.60) (USD 104.50 vs. USD 169.80), but there was not an obvious increase for urban patients. As co-MCD patients living in urban areas have easier access to improved infrastructure, medical resources, educational opportunities, and better social support and welfare, they attach greater importance to mental health and may take timely and appropriate treatment measures when they have depressive symptoms [57]. Patients residing in rural areas usually suffer from a shortage of health care providers and the need for extended travel to health care facilities [58]. In order to reduce transportation costs, patients fail to seek timely medical care for mild depressive symptoms, leading to more use of expensive medical services when their condition deteriorates.

The increased economic burden caused by depressive symptoms highlights the importance of screening for depression among middle-aged and elderly CLD patients. Early screening for depressive symptoms among middle-aged and elderly CLD patients would reduce health care costs. Previous studies have also pointed to problems of underdiagnosis and insufficient treatment of depression in the COPD population, with only 27% to 33% of co-MCD patients receiving antidepressant therapy [20,43]. Therefore, routine screenings for depressive symptoms should be part of the clinical encounter in these care settings so that appropriate treatment or timely mental health service referrals can be provided to this population.

Disparity in access to health services is an enduring concern for health care planners and policymakers in China [59]. Over the past several decades, China has seen remarkable economic growth and improved health care. However, rural–urban differences in health care resources [60] and health care utilization [39] have been reported. In fact, financing for China’s health care institutions partially depends on local governments, which vary considerably in their financial capacities between well-developed urban areas and under-developed rural villages. The local governments’ financial investments in health should be shifted to rural medical institutions to ensure accessibility for rural patients. Our study suggests that depressive symptoms are primarily associated with increasing hospital care for patients with chronic lung diseases. Therefore, we can establish mental health counseling services in primary health service institutions to diagnose and treat depressive symptoms in a timely manner and prevent patients from developing severe cases. In addition, it is essential to strengthen the construction of psychiatric departments and improve the physio-psychological integrated health care service capacity of primary medical staff.

Our study was the first to use nationally representative population data to reveal the correlation between depressive symptoms and health care service utilization and expenditure in middle-aged and elderly Chinese patients with CLDs, and to explore the differences between urban and rural areas. There were several limitations in our study. First, the medical expenditure data in the CHARLS questionnaire were derived from resident recall; consequently, information bias and recall bias were inevitable. Second, our study only included self-reported CLD patients; some important indicators associated with CLDs, such as lung capacity and the Global Initiative for Chronic Obstructive Lung Disease (GOLD) stage, were not included in the study. The mechanism of depressive symptoms in elderly CLD patients needs to be studied to reduce the economic burden of disease in CLD patients. Finally, our study was a cross-sectional design that was unable to establish a causal relationship between CLDs and depressive symptoms. Follow-up studies and panel data analysis are needed to further study the impact of depressive symptoms on the economic burden of disease in patients with chronic lung diseases.

## 5. Conclusions

Overall, our study found that depressive symptoms were significantly associated with additional medical utilization and expenditure in middle-aged and elderly CLD patients. In addition, the severity of depressive symptoms was associated with higher medical costs for patients with CLDs, especially rural patients. Routine screenings for depressive symptoms should be part of the clinical encounter in CLD patients’ care settings so that appropriate treatment and timely mental health service referrals can be provided to this population. Local governments’ financial investments in health should be tilted towards rural medical institutions to ensure accessibility for rural patients and narrow the health gap.

## Figures and Tables

**Table 1 ijerph-19-07849-t001:** Demographic characteristics of different depressive statuses among middle-aged and elderly people with CLDs.

Patient Variables	Non-co-MCDsN (%)	Lower co-MCDs N (%)	Higher co-MCDs N (%)	*p*
	N = 1122	N = 570	N = 142	
Gender				<0.001
Male	666 (59.4)	293 (51.4)	44 (31.0)	
Female	456 (40.6)	277 (48.6)	98 (69.0)	
Age				<0.001
45–54	216 (19.3)	119 (20.9)	27 (19.0)	
55–64	325 (29.0)	211 (37.0)	47 (33.1)	
65–74	357 (31.8)	178 (31.2)	58 (40.8)	
≥75	224 (20.0)	62 (10.9)	10 (7.0)	
Address				0.024
Urban	324 (28.9)	150 (26.3)	26 (18.3)	
Rural	798 (71.1)	420 (73.7)	116 (81.7)	
Marital status				0.016
Married	919 (81.9)	472 (82.8)	103 (72.5)	
Separated/Divorced	203 (18.1)	98 (17.2)	39 (27.5)	
Education level				0.004
Primary school and below	758 (67.6)	388 (68.1)	114 (80.3)	
Middle school	228 (20.3)	126 (22.1)	23 (16.2)	
High school and above	136 (12.1)	56 (9.8)	5 (3.5)	
Smoking				0.019
Smoker	338 (30.1)	173 (30.4)	38 (36.8)	
Former smoker	283 (25.2)	127 (22.3)	17 (12.0)	
Never smoker	501 (44.7)	270 (47.4)	87 (61.3)	
Medical insurance				0.413
Urban employee	184 (16.4)	72 (12.6)	8 (5.6)	
New cooperative	710 (63.3)	395 (69.3)	109 (76.8)	
Urban and rural resident	131 (11.7)	68 (11.9)	19 (13.4)	
Urban residents and others	63 (5.6)	25 (4.4)	3 (2.1)	
No medical insurance	34 (3.0)	10 (1.8)	3 (2.1)	
Occupation				0.015
Unemployed	462 (41.2)	213 (37.4)	63 (44.4)	
Retired	95 (8.5)	35 (6.1)	6 (4.2)	
Non-agricultural work	147 (13.1)	55 (9.6)	12 (8.5)	
Agricultural work	418 (37.3)	267 (46.8)	61 (43.0)	
Household monthly consumption				0.071
Lowest 20%	224 (20.0)	109 (19.1)	33 (23.2)	
Lower 20%	225 (20.1)	112 (19.6)	31 (21.8)	
Middle 20%	379 (33.8)	118 (20.7)	24 (16.9)	
Higher 20%	70 (6.2)	118 (20.7)	25 (17.6)	
Highest 20%	224 (20.0)	113 (19.8)	29 (20.4)	

Notes: CLDs, chronic lung diseases; non-co-MCDs, non-comorbid CLD and depressive symptoms; lower co-MCDs, lower comorbid CLD and depressive symptoms; higher co-MCDs, higher comorbid CLD and depressive symptoms.

**Table 2 ijerph-19-07849-t002:** The multiple analyses based on key variables in the incidence rate ratio of medical visits among middle-aged and elderly people with CLDs.

Variables	Incidence Rate Ratio	95% CI
Depressive (ref: Non-co-MCDs)		
Lower co-MCDs	1.30 ***	1.19–1.42
Higher co-MCDs	1.29 ***	1.12–1.49
Gender (ref: Male)		
Female	1.04	0.93–1.16
Age (ref: 45–54 years)		
55–64	0.94	0.84–1.06
65–74	0.91	0.81–1.03
≥75	0.98	0.84–1.15
Address (ref: Urban)		
Rural	1.13 **	1.01–1.26
Marital status (ref: Separated/Divorced)		
Married	1.07	0.96–1.20
Education level (ref: Primary school and below)		
Middle school	0.99	0.89–1.10
High school and above	1.07	0.93–1.24
Smoking (ref: Never smoker)		
Smoker	1.09	0.97–1.22
Former smoker	1.06	0.94–1.19
Medical insurance (ref: No medical insurance)		
Urban employee	1.04	0.77–1.39
New cooperative	0.99	0.74–1.29
Urban and rural resident	0.91	0.69–1.21
Urban resident and others	1.21	0.87–1.67
Occupation (ref: Unemployed)		
Retired	0.81 **	0.68–0.95
Non-agricultural work	1.00	0.87–1.16
Agricultural work	0.97 **	0.86–0.99
Household monthly consumption (ref: Lowest 20%)		
Lower 20%	0.85 **	0.75–0.97
Middle 20%	0.93	0.83–1.06
Higher 20%	0.97	0.86–1.08
Highest 20%	0.85 **	0.74–0.99

Notes: CI, confidence interval; CLDs, chronic lung diseases; non-co-MCDs, non-comorbid CLD and depressive symptoms; lower co-MCDs, lower comorbid CLD and depressive symptoms; higher co-MCDs, higher comorbid CLD and depressive symptoms. **: *p* < 0.05, ***: *p* < 0.001.

**Table 3 ijerph-19-07849-t003:** Additional medical utilization and expenditure in urban and rural areas caused by depressive symptoms among middle-aged and elderly people with CLDs.

	Urban				Rural			
	Lower Co-MCDs ^(2)^		Higher Co-MCDs		Lower Co-MCDs		Higher Co-MCDs	
	Marginal	95% CI ^(4)^	Marginal	95% CI	Marginal	95% CI	Marginal	95% CI
Health care utilization								
Total visits ^(1)^	0.6	−0.6–1.8	0.32	−2.1–2.7	2.5 ***	1.6–3.3	1.9 **	0.8–3.1
Outpatient	0.5	−0.2–1.2	1.0 **	0.4–1.6	−0.1	−0.9–0.8	−0.7	−2.4–1.0
Inpatient	0.02	−0.06–0.09	0.23 ***	0.10–0.35	0.04	−0.03–0.12	0.14 **	0.04–0.25
Self-medication	0.1 ^(3)^ **	0.03–0.12	0.04	−0.06–0.15	0.03 **	0.0–0.06	0.1 **	0.0–0.1
Health care expenditure								
Total costs	4908.1 **	1838.5–7977.7	4704.0 ***	2317.5–7090.5	1169.9 **	281.0–2058.7	2140.2 ***	1176.2–3104.2
Outpatient	812.3 **	82.7–1541.8	1651.2	254.5–3047.9	51.1	−496.6–598.7	696.5	−0.15–1393.1
Inpatient	2194.6 **	566.6–3822.7	2083.7	729.5–3437.9	690.9 **	259.3–1122.7	1122.6 ***	617.2–1627.9
Self-medication	582.9 **	96.7–1069.3	981.2	394.6–1567.8	495.9 ***	236.3–755.6	564.9 ***	254.3–875.6

Notes: (1) Robust regressions adjusted by gender, age, marital status, education level, smoking status, type of medical insurance, occupation, and household monthly consumption levels were used to get the results in all models. (2) Study groups were based on the scores on the CESD-10: non-co-MCD patients (score < 10), lower co-MCD patients (score 10 *≤* X *≤* 20), and higher co-MCD patients (score 21 *≤* X *≤* 30). (3) **: *p* < 0.05, ***: *p* < 0.001. (4) CI, confidence interval; CLDs, chronic lung diseases; lower co-MCDs, lower comorbid CLD and depressive symptoms; higher co-MCDs, higher comorbid CLD and depressive symptoms.

## Data Availability

The datasets generated and/or analyzed during the current study are available in the CHARLS repository, http://charls.pku.edu.cn/index/en.html (accessed on 3 September 2021).

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
