# Peer review of "The Additional Medical Expenditure Caused by Depressive Symptoms among Middle-Aged and Elderly Patients with Chronic Lung Diseases in China"

_ijerph, 2022, doi:10.3390/ijerph19137849_

Round 1

Reviewer 1 Report

I appreciate the opportunity to read this article. The issue of additional medical expenses for depressive symptoms is very relevant.

The paper is well written and synthetic, which strengthens the understanding of the model, results and discussion.

As suggestions for improvement:

- Clarify the hypothesis, objectives and method in the abstract and introduction.

- Comment briefly on some works that have addressed similar issues to compare results.

Author Response

Thanks very much for taking your time to review this manuscript. I really appreciate all your comments and suggestions! Please see the attachment to view my itemized response.

Reviewer 2 Report

This paper is an important addition to the evidence base in health-related economic costs.  CLD is high on the list of health-related causes of increasing health costs. The issue of depression associated with CLD and COPD imposes a high economic burden on society, therefore the topic of the study is timely and of scientific importance. 

The research methodology is clearly presented, results are easy to understand. However, I would recommend focusing on the results presented in Table 2 and interpreting what is the contribution of all variables tested, especially smoking status to marginal health expenditure.

Author Response

(The authors gave the same response as above.)

Reviewer 3 Report

Summary

The manuscript outlines research to investigate the medical utilisation and additional expenditure of depressive symptoms in middle-aged and elderly people with CLD in China, and then to assess urban and rural differences in economic impact.

The authors use data derived from the 2018 China Health and Retirement Longitudinal Study (CHARLS) database. Specifically, the panel includes 1,834 participants, including 1,122 (61.2%) with non-co-morbid CLD, 570 (31.1%) with lower co-MCD and 142 (7.7%) with higher co-MCD. For each patient, the authors estimate visit times and costs for outpatient, inpatient and self-medication.

The Authors use a zero-inflation negative binomial model to measure the relationship between each variable and the number of visits through the incidence ratio (IRR). In addition, the authors use a generalised linear model (GLM) with log-link and gamma distribution to estimate incremental medical costs.

The authors suggest that depressive symptoms are significantly associated with higher healthcare utilisation and expenditure among CLD patients. Based on the average economic loss of CLD patients of CNY 8,375.2 (USD 1,267.0), the additional medical costs caused by depressive symptoms increase by 20.0 % in lower CLD patients and 69.2 % in higher CLD patients.

Comments

The authors illustrate an interesting research on utilisation and additional medical expenditure related to depressive symptoms in middle-aged and elderly persons with CLD in China.

Overall, the research methodology is well structured, but it is sometimes difficult to link the analyses illustrated in the main text to the data presented in Tables 1-3. Consequently, the authors are invited to better articulate sections 2 and 3 by adding methodological content. 

In particular:

- The P-values reported in the last column of Table 1 seem to refer to the chi-square test introduced in row 143. Since it is not stated what is being tested, the Authors are invited to expand on lines 142 and 143 by mentioning the null hypothesis of the test (which is in any case a standard test), or at least a reference; in particular, it is not clear whether the test considers non-co-MCDs vs (lower + higher co-MCDs) or something else;

- the three percentages in lines 160-163 seem to be associated with the data shown in the first row of Table 1 (N). Authors may add a reference in the main text (e.g. see Table 1) or at least a footnote;

- even if it is a standard statistic, authors are invited to explain in the main text how they calculated the IRR (medical visits on patients with characteristic X/medical visits on patients without characteristic X, usually the reference group is indicated in brackets).

- The authors are invited to add in the main text the equations of the regression models they estimated to obtain the output shown in Table 3.

Minor Comments

Authors are invited to clarify the concept of "hukou" (line 140) at least in a footnote

The article is well-written, but some propositions are not well-structured. Consequently, I suggest to send the manuscript for (a light) proofreading.

Example1: Previous studies have proved depressive symptoms can affect the treatment and exacerbations of CLDs patients [42, 43]. But limited information is available on the medical economic burden effects of depressive symptoms in CLDs patients, particularly its differences in urban and rural status.

Why the dot after "patients"? A comma would be more appropriate, or "however" could be used instead of "but".

Example 2: In order to observe the disparity of economic burden effects of different depression degrees among CLDs patients, our study divided depressive symptoms into two levels, defined as mild to moderate depressive symptoms (with a score of 10≤X 110 ≤20) and severe depressive symptoms (with a score of 21≤X≤30). Corresponding lower co-morbid CLDs and depressive symptoms patients (co-MCDs patients) and higher co-MCDs patients.

Why the dot after "symptoms"? It may be more appropriate to sustitute the dot with a comma, and then rephrase as "identifying a group of lower... and a group of ...."

Author Response

(The authors gave the same response as above.)
